# Targeting Host Defense System and Rescuing Compromised Mitochondria to Increase Tolerance against Pathogens by Melatonin May Impact Outcome of Deadly Virus Infection Pertinent to COVID-19

**DOI:** 10.3390/molecules25194410

**Published:** 2020-09-25

**Authors:** Dun-Xian Tan, Ruediger Hardeland

**Affiliations:** 1S.T. Bio-Life, San Antonio, TX 78240, USA; 2Johann Friedrich Blumenbach Institute of Zoology and Anthropology, University of Göttingen, 37073 Göttingen, Germany; rhardel@gwdg.de

**Keywords:** melatonin, mitochondria, virus infection, innate immunity, MAVS, COVID-19, glycolysis

## Abstract

Fighting infectious diseases, particularly viral infections, is a demanding task for human health. Targeting the pathogens or targeting the host are different strategies, but with an identical purpose, i.e., to curb the pathogen’s spreading and cure the illness. It appears that targeting a host to increase tolerance against pathogens can be of substantial advantage and is a strategy used in evolution. Practically, it has a broader protective spectrum than that of only targeting the specific pathogens, which differ in terms of susceptibility. Methods for host targeting applied in one pandemic can even be effective for upcoming pandemics with different pathogens. This is even more urgent if we consider the possible concomitance of two respiratory diseases with potential multi-organ afflictions such as Coronavirus disease 2019 (COVID-19) and seasonal flu. Melatonin is a molecule that can enhance the host’s tolerance against pathogen invasions. Due to its antioxidant, anti-inflammatory, and immunoregulatory activities, melatonin has the capacity to reduce the severity and mortality of deadly virus infections including COVID-19. Melatonin is synthesized and functions in mitochondria, which play a critical role in viral infections. Not surprisingly, melatonin synthesis can become a target of viral strategies that manipulate the mitochondrial status. For example, a viral infection can switch energy metabolism from respiration to widely anaerobic glycolysis even if plenty of oxygen is available (the Warburg effect) when the host cell cannot generate acetyl-coenzyme A, a metabolite required for melatonin biosynthesis. Under some conditions, including aging, gender, predisposed health conditions, already compromised mitochondria, when exposed to further viral challenges, lose their capacity for producing sufficient amounts of melatonin. This leads to a reduced support of mitochondrial functions and makes these individuals more vulnerable to infectious diseases. Thus, the maintenance of mitochondrial function by melatonin supplementation can be expected to generate beneficial effects on the outcome of viral infectious diseases, particularly COVID-19.

## 1. Introduction

During the history of evolution, the battles of hosts and invading pathogens (parasites, bacteria and viruses) have never ceased and there are no absolute winners between them [1]. Coevolution and coexistence offer the only solution to these battles. Humans are the only species trying to dominate this battle and have achieved partial success. For example, in 1980, the World Health Assembly declared smallpox eradication globally. Humans have also made the measles, mumps, rubella, and varicella become vaccine-preventable illnesses [2]. However, we are still afflicted by many other pathogens, bacteria, as well as viruses. The influenza virus infection is a striking example of a persisting problem. In 1918, we lost tens of millions of lives during the Spanish flu pandemic. Till now, influenza virus infections cause 3–5 million severe cases and thousands of deaths globally each year. The high variability of influenza viruses and the emergence of new subtypes require an ongoing, but often incomplete, adaptation of vaccines [3]. In addition to flu, we have experienced several pandemics and epidemics of high concern that were caused by different viruses in less than two decades. These include the 2002 SARS, 2012 MERS, 2014 Ebola, and the current 2019 COVID-19. The impact of COVID-19 is present in nearly all countries and affects all races. This pandemic, for the first time in human history, has led to an almost global lockdown. With high likelihood, it will significantly change human behavior for times to come. This will and presumably has to include the way we handle co-existence with virus-bearing wildlife. Scientists and physicians have worked hard against the clock to identify the effective treatments as quick as possible by repurposing the existing medicines, in addition to vaccine development. This has proven to be a very difficult task. To date, no real breakthrough has been achieved by antiviral strategies in terms of reducing mortality from COVID-19. Some moderate success has been obtained by supportive treatments with anticoagulants [4,5,6] or anti-inflammatory agents, such as the IL-6 receptor blocker tocilizumab [7,8,9]. This situation is not totally surprising with regard to the history of anti-viral drug development. For decades of efforts, we have failed to develop anti-influenza medicines that are effective against the yearly varying subtypes. Vaccines against many other viruses are also missing [10]. The reasons for this are multiple, but the increase in the resistance of viruses to existing antibodies or other drugs due to their high mutation rates and serious side effects of the medicine to the hosts represent the main obstacles [11]. These drawbacks should force us to rethink the strategies that we have taken against virus infections. Whether we should mainly target a virus per se as we have done for decades or we should concentrate to the host as the therapeutic target may be the decisive question. Currently, the majority of efforts are focused on curtailing virus spreading, e.g., by blocking its entry into cells, inhibiting its replication, reducing its assembly. Notably, all these approaches are depending on synthetic compounds [12,13]. On the other hand, the blood from convalescent COVID-19 patients which contains naturally occurring antibodies against SARS-CoV-2 has also been used to stop virus invasion and development [14,15]. In contrast, few therapies have targeted the hosts, i.e., to modify the host’s defense strategy. From an evolutionary point of view, modification of host defense including enhancement of the tolerance to pathogen invasion seems more efficient than that of directly targeting the pathogens. With the exception of humans, host species have no means to selectively target pathogens using synthetic compounds, but they are able to modify their defense mechanisms. They survive and thrive under the pressures of different virus attacks. A good example is that of bats, which harbor a variety of viruses, some of which are deadly for other species including humans, but are harmless to bats [16]. The strategy of the bats against these potentially deadly viruses is to increase their tolerance by downregulating the innate immunity and inflammatory reaction rather than to enhance their defense against these viruses. This strategy is successful and has been acquired during evolution by gene mutations [17]. In addition, their tolerance to viruses has also been attributed to their higher levels of melatonin production compared to humans [18], a difference that may more favorably balance their immune responses. Melatonin is a naturally occurring molecule assumed to be present in every species [19]. Due to its antioxidant, anti-inflammatory, as well as other properties [20,21,22,23,24,25], it has been speculated to be a promising molecule for treating COVID-19 patients [18,26,27,28,29,30,31,32,33,34,35,36,37,38,39,40,41,42,43,44,45,46,47], and also other deadly viral infectious diseases [48,49,50,51]. A small scale clinical trial has also shown promising results [52]. The major activities of melatonin are not directed towards the virus per se and, thus, it does not kill the virus, but rather buffers the immune response of the host and increases the tolerance of the host to the virus infection [26]. Additional assumptions that melatonin may help to block virus entry into cells have also been hypothesized [28,42]. Regardless of this possibility, melatonin seems to reduce the severity and mortality for COVID-19. Its efficacy in other deadly viral diseases has been recently summarized [26]. We know that virus infections are self-limiting diseases if the patients have sufficient time for developing antibodies before the host organism has lost control. Most of the time, disease severity or mortality of the victim are not caused by direct viral cytotoxicity, but result from the manifestation of the host’s immune response. Sometimes, this is referred to as the cytokine storm, which unproportionally magnifies the signal of virus invasion and results in destructive inflammatory responses [26]. Thus, downregulation of the innate immune response, as in the case of bats, is a feasible strategy to increase the tolerance of the host. In addition, the strategy of targeting the host defense system has the advantage of being more widely applicable in different pathogen infections than only targeting a single pathogen by specific antiviral drugs and/or vaccines which would lose their protective effects in the inevitably upcoming next pandemic caused by a different pathogen. If we consider the potential convergency of two respiratory diseases such as COVID-19 and seasonal flu, this is highly expected with the winter approaching in the northern hemisphere [53]. This convergency would increase the severity of the symptoms [54]. Thus, the strategy of targeting the host’s defense is more reasonable than only targeting pathogens specifically. An established therapy to modify the host defense mechanisms will probably yield a similar efficacy against different pathogens and is readily available in the next pandemic or concomitance of more than one infectious disease. This is exactly what we expect melatonin will do. Since melatonin modifies host defense mechanisms, it will provide broad protection against different pathogen invasions and also preserve the precious time for the patients to develop effective antibodies required for final recovery [26]. The pathogen-related versatility of melatonin as well as its extremely favorable tolerability at the highest doses [55] are unmatched properties relative to other compounds. In this review, we will focus on the novel aspects as to how melatonin serves as an immune buffer to improve tolerance of the host against virus infections. Since this action of melatonin is mitochondria-associated, the associated roles of melatonin and mitochondria in the host defense against virus infections will be also reviewed.

## 2. Roles of Mitochondria in Viral Infection

Mitochondria are unique organelles in eukaryotic cells. The origin of these organelles can be traced back to α-proteobacteria. The theory of endosymbiosis [56] suggests that, 1.5–2 billion years ago, α-proteobacteria intruded the ancestors of eukaryotes. These bacteria can carry out aerobic respiration and, by means of an electron transport chain that builds up a proton potential, generate much more ATP for the host’s use than possible by glycolysis. As a result, hosts and intruders developed an endosymbiotic relationship. Since then, these α-proteobacteria underwent an evolution to mitochondria [57]. The primary function of mitochondria is, from its beginning to the present, that of energy metabolism, in particular, ATP production. During evolution, these organelles have acquired several additionally important functions, by which they participate, e.g., in cellular proliferative signaling, apoptotic control, fat metabolism, calcium homeostasis, aging, and innate immune responses with relevance to virus infections, but also autoimmune diseases [58,59]. Mitochondrial malfunction is crucial to numerous diseases referred to as ‘mitochondrial diseases’, which include neurodegeneration, myopathies, nephropathies, endocrinopathies, and cancer [60,61,62]. In this review, the focus is given to its innate immune regulatory function in virus infection.

As bacterial descendants, mitochondria still contain many substances that may be interpreted by the host as being foreign. Under normal, healthy conditions, this is prevented by the mitochondrial membranes, which avoids innate immune surveillance. These substances include the phospholipid cardiolipin, *N*-formyl peptides (n-fp), mitochondrial DNA (mtDNA) and its breakdown or modification products. Even metabolites that normally appear in the cytosol can induce innate immune responses. This has been shown for several metabolites of the citric acid cycle, in particular, succinate [63] and ATP, when viral RNA is detected by RIG-1 (retinoic acid-inducible gene I) [64]. When these substances are released from the mitochondria either actively or passively, they will be exposed to the immune surveillance and detected by the pattern recognizing receptors as components of the damage associated molecular pattern (DAMP). Thus, these substances can initiate immune responses and inflammation by DAMP receptors in both immune and non-immune cells [65].

The immune regulatory effects of mitochondria have been extensively studied. These organelles play the pivotal immune regulatory role against viral infection [66]. A central part for such a role is a membrane-bound receptor known as mitochondrial antiviral-signaling protein (MAVS). MAVS is a 540-amino acid protein encoded by the nuclear genome and mainly anchored at the mitochondrial outer membrane with its C-terminal transmembrane (TM) domain. Its N-terminal caspase recruitment domain (CARD) serves as a receptor for both RIG-I and melanoma differentiation-associated gene 5 (MDA5) ligands. RIG-I and MDA5 belong to the retinoic acid-inducible gene-I-like receptors (RLRs) family. They are cytosolic PAMP (pathogen associated molecular pattern) recognition receptors, particularly for virus RNA [67]. Once RIG-I and MDA5 detect virus RNA, they interact with the CARD of MAVS to initiate antiviral and inflammatory cascades. The down-stream signaling pathway of MAVS includes activation of the transcription factors interferon regulatory factors 3 and 7 (IRF3/7) and their translocation to the nucleus. This process upregulates the expression of IFNs and IFN-stimulated genes (ISGs) to inhibit viral replication and transmission. Activated MAVS also induces activation and nuclear translocation of NF-κB (nuclear transcription factor-κB), which upregulates the expression of various proinflammatory factor genes. On the other hand, viruses can also attack the MAVS pathway to escape from the host’s antiviral immune response. This can occur by targeting MAVS directly, e.g., via enhancing its cleavage, or by interfering with other components involved in this pathway. For example, the open reading frame 9b (ORF-9b) of SARS-CoV-1 catalyzes a K48-linked ubiquitination and degradation of the MAVS signalosome [68], whereas the HCV (hepatitis C virus) protease NS3/4A cleaves MAVS off the mitochondria, and HAV (hepatitis A virus) uses a stable, catalytically active polyprotein processing intermediate to target MAVS for proteolysis [69].

In addition to the MAVS pathway, under the stress of virus infection, mitochondria also release their mitochondrial DNA (mtDNA) or other substances into the cytosol or to the extracellular space, where they act as mitochondrial alarmins [70]. How these mitochondrial alarmins are released after virus infection is not yet fully understood. This release may be regulated by an altered mitochondrial membrane potential, its associated changes in the mitochondrial permeability transition pore (mtPTP) and mitochondrial outer membrane permeabilization (MOMP), respectively [71,72]. Upon their release, the endosome membrane-bound toll like receptor 9 (TLR-9) and cytosolic cGAS (cyclic GMP-AMP synthase) can recognize mtDNA as a DAMP [73]. Interestingly, based on the molecular structure of the TLR-9 variants, especially concerning their truncated N-terminals, it has been predicted that TLR9-B would localize to mitochondria to recognize mtDNA [74]. The biological consequences of this observation are currently unknown. Via TLR-9/MyD88 and cGAS/cGAMP/STING pathways, these DAMP-recognizing receptors upregulate the expression of IFN genes to initiate their antiviral activity. In addition, mtDNA serves as a ligand of the NLRP3 multiprotein complex [75], which is tethered by cardiolipin, when translocated to the outer membrane, to the mitochondria, where both ligands activate the NLRP3 inflammasome [76].

The innate immune response to a virus infection also involves aspects of mitochondrial metabolism. ROS, which are byproducts of mitochondrial metabolism, serve as molecular signals, as they induce aggregation of MAVS on the mitochondrial outer membrane to initiate the IFN response [77], and can also counteract virus replication [78]. However, excessive ROS production is associated with viral pathogenesis, as shown for infections by hepatitis C and B in liver and by SARS-CoV-2 in lungs [79]. The delicate balance of ROS production and its quenching is important for virus infection and spreading. Virus infection can result in a switch of mitochondrial oxidative phosphorylation metabolism to anaerobic glycolysis [80] which is referred to as the Warburg effect [81]. Anaerobic glycolysis favors viral replication. The viruses divert the glycolytic carbon for biosynthetic means, and increase glutamine utilization for virus-specific purposes [82]. The glycolytic metabolism also favors macrophage polarization from the anti-inflammatory type M2 to the proinflammatory type M1. M1 macrophages rely on anaerobic glycolysis for energy production, whereas M2 macrophages generate ATP via the TCA cycle and the respiratory chain [83]. The predominating M1 macrophages secrete a variety of proinflammatory cytokines and chemokines, which is manifested as the cytokine storm observed in SARS-CoV-2 [84]. These alterations facilitate destructive inflammatory reactions that are typical for many viral infections. In conclusion, mitochondria regulate via multiple pathways the innate immune and inflammatory responses to virus infections. The MAVS pathway plays a central role and coordinates this regulation. On the other hand, viruses also manage to interfere with MAVS signaling and mitochondrial metabolism, in order to avoid the host’s innate immune defense. These actions often lead to mitochondrial damage, cytokine storm, and destructive inflammation. Thus, functional mitochondria and a balanced innate immune response of the host are critical for the outcome of deadly virus infections including SARS-CoV-2. The roles of mitochondria in virus infection are summarized in Figure 1.

## 3. Mitochondria and Melatonin

As mentioned above, mitochondria have evolved from α-proteobacteria via endosymbiosis. Interestingly, these bacteria have already had the capacity to synthesize melatonin [85,86]. Based on this evidence, it has been hypothesized that the melatonin biosynthetic capacity of these bacteria has been preserved by the modern mitochondria [87]. Indeed, melatonin synthetic enzymes have been identified in mitochondria of organisms including animals and plants [88,89,90,91]. It is reported that the mitochondrial site for melatonin biosynthesis is exclusively confined to the matrix [90]. From the point of view of substrate availability, the mitochondrial matrix is the ideal place for melatonin production. Acetyl coenzyme A (acetyl-CoA), a cofactor for melatonin synthesis, is mainly generated in mitochondria with a concentration fitting well to the K_m_ of AANAT [92], the rate-limiting enzyme of melatonin synthesis. Another important substrate for melatonin synthesis is *S*-adenosylmethionine (SAM), which provides the methyl group for melatonin synthesis catalyzed by the acetylserotonin methyltransferase (ASMT). SAM has a stable concentration in mitochondria compared to other cellular compartments [93]. Thus, substantially high melatonin is detected in mitochondria relative to cytosol [94]. High melatonin is required for maximizing the mitochondrial functions. The primary function of mitochondria is energy production by oxidative phosphorylation. During this process, some electrons will inevitably leak from the electron transport chain and interact with oxygen to generate superoxide, the most abundant ROS, which can be converted to other reactive species. Under most conditions, the levels of ROS are delicately controlled by the mitochondrial antioxidant system and are maintained within a certain range suitable for serving as signaling molecules. Otherwise, excessive ROS levels result in oxidative stress and lead to cell and tissue damage. Melatonin is an important member of the antioxidant system [95]. It is a potent direct free radical scavenger and also acts as an indirect antioxidant by enhancing the activities of antioxidant enzymes including mitochondrial SODs, catalase and glutathione reductase [96,97]. Importantly, it has the potential of reducing mitochondrial ROS generation by upregulating the activity of UCP1 (uncoupling protein 1) in brown adipose tissue [98,99] and UCP 3 in cardiomyocytes [100]. The capability of decreasing ROS formation, which is based on multiple mechanisms, has been referred to as the ROS avoidance activity of melatonin [101]. All of these actions render its protective effects on ROS-induced oxidative stress in mitochondria. Indeed, many studies have reported the protective effects of melatonin on tissue injuries caused by oxidative stress. These include ischemia/reperfusion induced injuries of brain, lungs, heart, intestine, testes and skeletal muscle as well as hypoxia-caused tissue and organ damages [102,103,104,105,106,107]. The protective effects of melatonin are largely associated with its activity on mitochondria. For example, melatonin increases the activity of mitochondrial complexes 1, 3, and 4 [108,109], balances the mitochondrial membrane potential [110], reduces the duration of opening of the mitochondrial membrane permeability transition pore (mtPTP) [111], and maintains the ATP production [112]. Melatonin is classified as the mitochondrially targeted antioxidant [113]. Thus, reduced melatonin levels are associated with several diseases including neurodegenerative diseases (Alzheimer’s disease, Parkinson’s disease, and amyotrophic lateral sclerosis), heart disease, breast and prostate cancers, multiple ovarian cysts, diabetes, and systemic metabolic disorders [114,115,116,117]. Conversely, melatonin treatment has beneficial effects on most of these diseases [118,119,120]. Mitochondria are, aided by the host cell, self-renewing organelles and frequently undergo processes of fission and fusion. The periodically occurring fission and fusion cycles of mitochondria are referred to as mitochondrial dynamics. Generally, fusion expands the size of mitochondria and enhances their function. Fission increases mitochondrial numbers and is necessary for the replication of the cells. However, in post-mitotic cells, a function of fission is to separate dysfunctional segments of the mitochondria from the normal ones. The separated dysfunctional segments enter an autophagic process known as mitophagy [121]. This implies that fission is to some degree associated with mitochondrial injury in post-mitotic cells. It has been reported that fission is caused by a reduced mitochondrial membrane potential in the injured area [122]. Mitochondrial membrane-bound proteins, including mitochondrial fission protein 1 (Fis1), mitochondrial fission factor (Mff), mitochondrial elongation factor 1 (MIEF1/MiD51), and mitochondrial elongation factor 2 (MIEF2/MiD49), act as receptors of dynamin-related protein 1 (Drp1) and these receptors can sense alterations of the mitochondrial membrane potential and recruit the cytosolic Drp1, a GTPase, to execute this fission process [123,124]. Melatonin, most of the time, promotes the mitochondrial fusion and reduces fission, thus keeping the mitochondria in their functional state [113]. The primary mechanism by which melatonin reduces mitochondrial fission is to preserve the mitochondrial membrane potential by inhibition of mtPTP. mtPTP seems to have several heterogeneous structures, which include pores or channels that are located in both the mitochondrial outer and inner membranes. Long-lasting openings cause leakages between the matrix, intermembrane space, and cytosol, and therefore lead to the collapse of the mitochondrial membrane potential. The structures of mtPTP are not fully clarified. The voltage dependent anion channel 1 (VDAC1), translocator protein (TSPO) (previously known as the peripheral benzodiazepine receptor) located in the mitochondrial outer membrane, cyclophilin-D in the mitochondrial matrix, ADP/ATP carrier (AAC) in the inner membrane (also known as adenine nucleotide translocase, (ANT), and more are assumed to be mtPTP components. Cyclophilin-D is believed to be a key regulatory factor, but not part of the pore [125,126]. Another regulatory mechanism concerns AAC, which binds cardiolipin to form a closed state of the pore. Under the increased production of ROS, the oxidation of the bond between cardiolipin and AAC causes a conformational change of AAC that opens the pore. As a mitochondrially targeted antioxidant, melatonin protects cardiolipin from oxidation and inhibits the AAC-associated mtPTP opening. For VDAC1 associated mtPTP opening, melatonin activates AMPKα to blunt Drp1-dependent mitochondrial fission and prevents VDAC1 opening [127]. Melatonin inhibits the Ripk3-PGAM5-CypD signaling pathways to block the cyclophilin-D associated mtPTP opening [128]. The mtPTP opening is also associated with another important cellular event, namely apoptosis. Enduring mtPTP opening results in the release of cytochrome C into the cytosol, a signal that activates caspase 3 and initiates apoptosis. Many diseases and disorders are accompanied with excessive apoptosis. Melatonin is a molecule that inhibits apoptosis in different cells, tissues and organs under a variety of inducers. The main mechanism consists in the inhibition of mtPTP opening to prevent cytochrome C leakage [129,130]. Effects of melatonin on mitophagy have been described but are complicated and depend on the cell type and study conditions. Some studies have reported that melatonin promotes mitophagy [131,132,133,134], but others show inhibition [135,136]. If the data are carefully analyzed, it appears that mitophagy promotion is the predominant aspect of melatonin. In this context, it should be noted that mitophagy can be either favorable or deleterious, In the former case, it cleans the cell from garbage, but in the latter case, extensive mitophagy can lead to severe depletion of peripheral mitochondria, which causes, e.g., losses of neuronal connectivity [137,138,139]. Mitophagy inhibition may not be a direct action of melatonin, but may indirectly result from the protective effects of melatonin on the mitochondria. When the structure and function of mitochondria are preserved by melatonin, the mitochondrial fission should be reduced, and thus less mitophagy can be expected. This conclusion requires well-designed studies to prove it. It seems that melatonin is essential for the preservation of functional mitochondria in all cells. The relationship of melatonin and mitochondria has been extensively reviewed [113] and is illustrated in Figure 2.

## 4. Melatonin Synthesis: A Target of Virus Infection

As mentioned above, melatonin is a phylogenetically ancient molecule. In almost all organisms studied, its primary function is that of serving as an antioxidant and free radical scavenger [140]. In eukaryotic cells, this is associated with detoxifying and balancing the levels of ROS mainly generated in the mitochondria. The other functions of melatonin are acquired during evolution [19], and this includes its immune regulatory function in hosts. It seems that melatonin is selected as a common signal of environmental and internal stresses by both unicellular and multicellular organisms from bacteria to mammals. For example, when a unicellular organism, the dinoflagellate *Lingulodinium*, was exposed to the cold stress, it increased its melatonin production by orders of magnitude [141]. Similar observations have been reported in fungi [142]. In animals, stress-related melatonin synthesis is referred to as stress-induced and stress-released phenomena [143]. This phenomenon has been extensively studied also in plants. Not only does abiotic stress, such as cold, heat, drought, chemical, and metal pollutants, stimulate plant’s melatonin production. Most importantly, biotic stress by virus, bacterial, and fungal infections also enhances melatonin biosynthesis [144,145]. The infection by microbes leads to gene upregulation of the melatonin synthetic enzymes and enhanced levels of melatonin. In turn, this increases the tolerance of hosts to the pathogens by regulating the innate immune response. This stress-induced and stress-released melatonin phenomenon is extremely important for organisms’ survival under stressful conditions. A compromise of this ability will result in catastrophic consequences for the organisms, especially under virus infection. The host and the pathogens respond differently to the universal signal of increased melatonin levels. This signal usually guides the host cells to balance their innate immune response by curbing the overshooting innate immune action and inflammatory reaction as well as promoting their adaptive immune reaction. This will significantly increase the tolerance of the host against the pathogen invasion and increase the survival chance of the host under the deadly virus infection. For the pathogens (parasites, bacteria, fungi and viruses), the high melatonin level raised by the host is a signal indicating an unfriendly environment and a signal for them to hold rapid replication. This has been observed in the in vitro studies of bacteria, fungi and virus [146]. To maximize their replications, these pathogens have to target the host’s melatonin synthetic system to bring the melatonin concentration down. It seems that targeting the melatonin biosynthesis to modify the host defense is a useful strategy for their rapid spreading inside of the host, e.g., when the bacteria *Candidatus Liberibacter asiaticus* infected insects *Diaphorina citri*. These bacteria attacked the host’s melatonin synthetic pathway by suppressing the gene expression of all melatonin synthetic enzymes including *TPH, AAAD, SNAT*, and *ASMT* and shut down melatonin production [147]. Another strategy taken by microbes including viruses is simply to deplete the precursor of melatonin synthesis, tryptophan, by activating indolamine 2,3-dioxygenase (IDO), which metabolizes tryptophan to kynurenine catabolites [148]. In mammals and some plants tested, IDO also metabolizes melatonin to AFMK [149,150,151]. Considering that mitochondria are major synthetic sites for melatonin production, targeting of mitochondria is a potential mechanism for viruses to lower melatonin production. There is no direct evidence to show this, but reports on viruses manipulating mitochondrial structure and function to alter the host’s innate immune responses are not scarce [152,153,154,155]. With regard to SARS-CoV-2, its impact on mitochondria just begins to emerge. It has been predicted that SARS-CoV-2 RNA is abundantly localized in host mitochondria [156] and this indicates that SARS-CoV-2 hijacks and modulates host cell’s mitochondrial function to viral advantage, for example, to suppress innate and adaptive immunity for rapid viral replication [157]. Mitochondrial dysfunction caused by the virus seems to contribute to the progression and severity of COVID-19 [158]. Any mitochondrial dysfunction will jeopardize melatonin production. In many cases, viruses direct the host cell metabolism to anaerobic glycolysis for their rapid replication while the cellular energy demand will be adapted to glutamine metabolism (glutaminolysis) [159,160]. A mechanism for this is that viruses infection triggers mitochondrial ROS production, which stabilizes hypoxia-inducible factor-1α (HIF-1α) and consequently promotes anaerobic glycolysis in COVID-19 infection [161], as known for other viruses. This process depletes acetyl-CoA for melatonin production as discussed above. In addition, melatonin *per se* has the capacity to switch energy metabolism from anaerobic glycolysis to mitochondrial oxidative metabolism, which has been observed in tumor cells and also in other cells by downregulating the expression of LDH and also reducing its activity [162]. As a free radical scavenger, melatonin inhibits the ROS-mediated activation of the HIF-1α/miR210/ISCU axis, a pathway which promotes anaerobic glycolysis [163]. Recently, Reiter et al. have hypothesized that melatonin may suppress the activity of pyruvate dehydrogenase kinase (PDHK) to activate the function of PDH and thus cause a switch from anaerobic glycolysis to mitochondrial oxidative metabolism in tumor cells and in mast cells infected by SARS-CoV-2 [164,165]. In addition, SARS-CoV-2 can spread into the brain, thereby dysregulating mitochondrial metabolism and causing neurological damage. This neuroinvasive capacity of SARS-CoV-2 is speculated to be suppressed by melatonin [43,166]. It is evident that the reduced melatonin formation caused by a manipulating virus will lead to a vicious cycle of melatonin production. Reduced salivary [167] and serum melatonin levels [168] have been observed in human immunodeficiency viruses (HIV) infected patients. The potential mechanisms of virus to attack mitochondrial melatonin synthesis are illustrated in the Figure 3. The outcomes of deadly virus infections may be based, to some degree, on the battle of host and pathogen, which affects mitochondrial melatonin production. On the one hand, the host cells are stimulated to upregulate their melatonin production by means of the “stress-stimulated and stress released melatonin synthetic mechanism”. On the other hand, a virus can bring melatonin synthesis down by the mechanisms mentioned above. This battle is rooted in the evolution. Under most conditions, the hosts win the battle and recover from virus infections. Unfortunately, some individuals lose this battle due to their already compromised mitochondrial functions, as occurs in elderly subjects, some males, or in persons with predisposed health conditions, which has become evident in COVID-19. These issues will discuss hereinafter.

## 5. The Outcomes of COVID-19 Are Potentially Linked to Melatonin Production

There are several risk factors for the severity and mortality of COVID-19 patients. These include, but are not limited to aging, gender, predisposed health conditions such as hypertension, obesity, cardiovascular diseases, diabetes, pregnancy, and other pathologies. These issues will be discussed in detail in the following parts of this section.

### 5.1. Aging

Aging is a major risk factor of outcomes for COVID-19 patients. Statistical analyses indicate that the relative number of elderly subjects contracting the virus is far higher than the proportion of elderly in China and South Korea, probably also in other countries [169]. However, the proportions between age groups are meanwhile changing in some countries, e.g., Germany, in which the relative number of infections is substantially increasing in the younger population, especially in the groups of 5–14, 15–34 and 35–59 years (Robert-Koch-Institut: COVID-19-Dashboard. http://arcgis.com/experience/47822a4c454480e823b17327b2bf1d4). However, one has to be aware that the percentage of infections is lifestyle-sensitive and less important than the percentage of severe disease progression, which is clearly highest in elderly patients. For example, 85.6% mortality occurred in COVID-19 patients older than 65 years in Spain [170].There are many reasons for this, for example, fragility of the older population, immunosenescence, which is associated with increased levels of proinflammatory cytokines and the problem of developing an immune-risk profile [171,172,173,174], and chronic low-grade inflammation (inflammaging) [91,119,171,173,174,175,176]. All of these are more or less associated reduced melatonin production in the aging population. The decline of melatonin production with age is a universal phenomenon in organisms. In humans, serum melatonin is mainly released from the pineal gland which only generates a small portion of total melatonin produced by our body. For example, the gut can produce several hundred folds more melatonin than the pineal gland [177]. However, the extrapineal melatonin is poorly released into the circulation under physiological conditions and does not substantially contribute to the serum melatonin [178], except for a post-prandial release that depends in its extent on the kind of food [179]. The serum melatonin largely reflects the function of the pineal gland, but this parameter has been used for decades to evaluate the body’s melatonin production. Serum melatonin also exhibits a circadian rhythm with a pronounced peak at night and baseline during the day. The meaningful measure for serum melatonin is its night-time concentration, whereas the day-time level is not considered to be of biological significance. Currently, it has been reported that the gene expression of melatonin synthetic enzymes in mitochondria of other cells such as in neurons does not exhibit a circadian rhythm. This observation raises a question on the accuracy of current melatonin measurements and the significance of this observation remains to be clarified in the future. Anyway, if we mentioned melatonin levels in this text, it usually implies the night time serum melatonin, if not otherwise specified. In humans, the highest melatonin production is in their prepuberty period. Until puberty, the pineal gland synthesizes melatonin without substantial age-dependent changes, and a decrease of the average nocturnal level in the course of childhood represents a “dilution effect” by the increase in body size [180]. After puberty, melatonin production begins to decline with age, which is seen in the serum levels. For example, in humans, at the age of 1–3 years, their nighttime serum melatonin is 329.5 ± 42.0 pg/mL. However, at the age of 70–90, this level is around 29.2 ± 6.1 pg/mL [181]. This is less than one tenth of the children’s level. There are several mechanisms to explain the reduced melatonin production in the elderly: (1) After the 60s, most of the individuals have developed a pineal gland calcification (pineal stone) [178]. The portion of pineal gland with calcification restricts its melatonin production. (2) The number of β-adrenergic receptors located in the pinealocytes or their sensitivity decline with age. This receptor is mainly responsible for the stimulation of pinealocytes to generate melatonin. (3) The gene expression of melatonin synthetic enzymes or their phosphorylation are downregulated with aging. (4) Neurodegeneration in the circadian master clock, the SCN, and/or its connections to the pineal gland lead to reduced circadian amplitudes and decreased nocturnal pineal stimulation [182]. (5). Melatonin consumption is increased with aging due to the increased ROS production with aging. Thus, the compromised melatonin production in the elderly may cause difficulties in coping with the stress of a SARS-CoV-2 infection, leading to the highest severity and mortality in this group.

### 5.2. Comorbidities

The predisposed health condition is another risk factor next to aging with relatively high severity and mortality for COVID-19 patients. These include obesity, hypertension, cardiovascular diseases and diabetes [183,184,185,186]. Compromised melatonin production or altered melatonin receptor function are one of the common etiologies of these disorders. In obesity, melatonin has been used to reduce body weight and adipose storage in animal studies for decades. Human trials have just caught up recently and have shown some promising signs [187,188]. The serum level of melatonin in obese subjects seems to have two different phases. In the early stage of obesity, the serum melatonin is elevated, and this elevation is considered as a protective mechanism to fight high free radicals in obesity [189]. However, in the case of extreme obesity, the melatonin level is significantly reduced compared to the controls [190]. Recently, it has been reported that the pineal gland size is significantly smaller in obese individuals than that in lean controls [191]. This observation also supports the melatonin deficiency in obese subjects. One of the mechanisms of melatonin in the case of obesity is to target the mitochondrial energy metabolism. Melatonin activates the uncoupling proteins 1 (UCP1) (in brown adipose tissue) and UCP3 (in other cells) located in mitochondrial inner membrane, which facilitates a proton backflow from the intermembrane space to the matrix, in the absence of ATP formation, generating heat and dissipating the extra energy. In this way, melatonin balances the energy metabolism and reduces mitochondrial ROS production. Melatonin deficiency in obese individuals has already jeopardized this process and resulted in elevated oxidative stress. Under the additional stress of the virus infection, especially, when melatonin synthesis is targeted by virus, the extremely low melatonin may contribute to a collapse of mitochondrial function and decreased tolerance of obese persons to the virus infection. Therefore, the use of melatonin has been suggested to improve the clinical outcome in COVID-19 patients with obesity and diabetes [192]. In cardiovascular disease and hypertension, patients usually suffer from melatonin deficiency and are prone to heart infarction or sudden death [116,193,194]. Sudden death related to a cardiac event often occurs in the early morning, 1–3 h after awakening [195,196], a phase during which the melatonin level rapidly drops from its peak to a low daytime level. During this period, vascular system and heart do not cope well with this sharp decrease. Thrombosis is a major culprit of sudden death, which is also observed in severely ill COVID-19 patients and fatal cases [197,198]. Vascular thrombosis in vital organs including heart, lung and brain leads to increased severity and mortality in COVID-19 patients. Complement and platelet activation, increased tissue factors and endothelial damage are considered as main causes of the COVID-19-associated vascular thrombosis. These changes can be interpreted as the consequence of overshooting inflammation due to hyperactivation of the innate immune system [4,5,6,7,8,9]. In addition to its anti-inflammatory actions, melatonin promotes the release of tissue factor pathway inhibitor (TFPI) from intracellular stores or from glycosyl phosphatidylinositol anchored TFPI present on endothelial membranes [152]. TFPI is a serine protease inhibitor that counteracts TF-induced coagulation through a Factor Xa-dependent feedback inhibition of Factor VIIa [199] and is particularly suitable for preventing thrombosis, especially with regard to inflammation-induced coagulation [200]. Platelet activation can be inhibited by melatonin, too, and has been interpreted as a consequence of reduced mitochondrial ROS [201,202]. However, endothelium–platelet interactions should not be overlooked in such interpretations, also with regard to endothelial NADPH oxidases, which are a main source of ROS in the vasculature [203]. Clinical studies have found that melatonin treatment reduces the peak thrombin generation in tetraplegic patients who are prone of venous thrombosis [204]. A further melatonin deficiency induced by the SARS-CoV-2 infection in patients with hypertension and heart diseases would inevitably result in the increase of vascular events, worsen symptoms and presumably increase mortality. Many studies have investigated the outcomes of diabetes with SARS-CoV-2 infection [205,206,207], but few have addressed the importance of melatonin in this context. Diabetes has a complicated relationship with melatonin. One the one hand, melatonin inhibits insulin secretion of the β-cells and increases the fasting glucose level [208]. On the other hand, melatonin increases the sensitivity of cells to insulin and, hence, can reduce the blood glucose level, under certain conditions [209,210]. This paradox may be resolved if we assume that the inhibition of insulin secretion by β-cells may preserve these cells from becoming exhausted under continuous stimulation by high glucose and may also reduce insulin resistance. This is consistent with the fact that melatonin treatment prevents β-cell damage induced by chemicals and glucotoxicity under in vivo and in vitro conditions [211,212,213]. Clinical studies also show that the serum melatonin levels are inversely correlated to insulin resistance [214]. These antidiabetic effects can, however, be compromised by prodiabetic increases in the expression of an MT_2_ melatonin receptor variant (the “G allele” carrying the SNP rs10830963), which disproportionally rises in homozygous carriers after the age of 40 years and strongly suppresses insulin secretion by β-cells [215,216]. Although a genetically based strong suppression of insulin secretion is, without any doubt, a diabetic effect, this does not tell much about the etiology of diabetes in carriers of unmutated MT_2_ receptor genes. Moreover, the necessity to distinguish between reduced insulin secretion and insulin resistance has been emphasized [217]. Our main point is that diabetes is associated with low melatonin levels [218,219,220,221] and that reduced melatonin may be regarded as a modifiable risk factor for diabetes within the general population [221]. This risk factor of low melatonin may magnify the symptoms of COVID-19 in diabetics. As mentioned above, the virus prefers anaerobic glycolysis rather than mitochondrial respiration for its replication and spreading. The constant high level of glucose in diabetes facilitates this purpose, especially under conditions of low melatonin, which weakens melatonin’s capacity to switch from anaerobic glycolysis to oxidative metabolism. The low melatonin level in diabetes will also decrease mitochondrial antiviral activity, facilitate inflammation, favor cellular apoptosis, and thus contribute to virus spreading and symptom manifestation of COVID-19 in these patients.

### 5.3. Gender

As in some other infectious diseases, males have been reported to be more prone to COVID-19 than females. In terms of infection rates, this not equally evident in all countries. In Germany, for instance, no substantial difference between genders has been detected in this regard, in whatever age group (Robert-Koch-Institut: COVID-19-Dashboard. http://arcgis.com/experience/47822a4c454480e823b17327b2bf1d4). However, the severity and mortality of the male COVID-19 patients are significantly higher in some statistics than those of females. A current meta-analysis showed that 66.6% of COVID-19 mortality concerns males, with a median age around 70 years [222]. The higher mortality of men than women is not completely explained by the higher prevalence of comorbidities in men [223]. Various other factors have been attributed to this gender difference of COVID-19. These attempts of explanation include the sex hormones, gene doses of X and Y chromosomes, gender-specific microbiota, mother lineage of mitochondria, etc. [224]. Here, the focus is given to the potential association of melatonin with the higher susceptibility of males to COVID-19. Melatonin is mainly synthesized in the mitochondria of presumably all cells. The status of mitochondria decides on the production of melatonin. It is hypothesized that the female’s mitochondria exhibit some advantages over the male’s ones due to the maternal transmission of mitochondria to both genders. This may have led to an evolutionary advantage of mitochondrial functionality in females. Although both genders receive the same mitochondria from their mother, the female cells may be more perfectly adapted to efficient interactions with mitochondria. For instance, the male’s mitochondria may be more easily subjected to stress-induced damage than those of females. Such phenomena of sex dimorphism that disfavor males are known in biology as the so-called mother’s curse and have been observed in the context of mitochondria [225]. Under normal conditions, the melatonin synthetic activity in male’s mitochondria can match the demand of the cells and this conforms with the observation that serum melatonin levels do not significantly differ between males and females, neither in animals [226] nor in humans [181]. However, males and females deviate in their responses toward stress regarding melatonin production. When males and females were challenged by thyrotrophin releasing hormone (TRH) injection, a significant increase in melatonin levels was only observed in females, but not in males [227]. This phenomenon was also observed in animal studies on stressful conditions. If fetuses are stressed during gestation by maternal undernutrition, the male offspring has, in the phase of prepuberty, much lower melatonin levels than the females. Correspondingly, males suffer more strongly from oxidative tissue damage than the females [228]. The higher melatonin levels in females are not associated with estrogen alterations, but represent a difference in stress response. The lower melatonin level of males may indicate that their mitochondria cannot equally respond to stress as females do. If this is applied to the SARS-CoV-2 infection, it may appear plausible why males suffer more frequently than females in this disease. The male’s inborne disadvantage concerning mitochondria (mother’s curse) implies a poorer resilience to extensive stress, with a reduced capacity to synthesize melatonin. The potential associations between mitochondrial function, melatonin action, and gender difference during SARS-CoV-2 infection have been addressed in the literature [224].

### 5.4. Children and Pregnant Women

As mentioned in Section 5.1, children have the highest melatonin levels and this may be related to their less severe symptoms and lowest mortality relative to other age groups in COVID-19 patients [229,230]. Pregnant women with COVID-19 are a special population with a mostly lower grade of severity, less enhanced inflammatory response and more favorable cell immunity than other individuals in a similar age group [231]. Many causes may contribute to their better resistance to COVID-19. An elevated melatonin level may be one of these reasons. Serum melatonin of pregnant women gradually increases in the course of gestation, especially during the third trimester of pregnancy, and reaches its peak at the time of delivery [232,233]. Since trophoblast [234] and placenta [235,236], as well as embryo [237,238] and stages of the fetus have the capacity to synthesize melatonin, the melatonin of pregnant women is probably derived from the mother, fetus, and placenta as a whole. Thus, pregnant women have a significantly elevated serum melatonin level, which is approximately twice as high as that of control subjects or even higher [181]. Whether the high level of melatonin is decisive for the resistance of pregnant women to COVID-19 deserves further investigation.

## 6. Conclusions

Entering the 21st century, one of the biggest challenges for public health seems to come from viral infectious diseases. During less than 20 years, the world has experienced several pandemics caused by viruses, including avian flu, SARS, MERS, Ebola, and COVID-19. To date, there are no convincingly effective specific antiviral medicines available and the development of vaccines is extremely time-consuming. This should prompt us to reconsider our strategies for combating viral infections. The decisive alternative is whether we continue to mainly target the pathogens, as we have done for the past several decades, or whether we should focus our efforts on primarily targeting the host. We think that targeting the host defense system to increase the tolerance is a better strategy than solely targeting the pathogens. This should not be misunderstood as an underappreciation of vaccine development. However, the time required until a new vaccine is approved and available as well as the necessity of developing, again and again, a new vaccine for every newly appearing virus are practical obstacles that should be overcome. A generalized strategy of strengthening the host’s defense and avoiding overshooting innate immune responses that lead to high-grade inflammation is clearly of advantage, is versatile for treating different diseases, and is readily available. In the natural world, bats have successfully proven the beneficial effect of this strategy. They survive loaded with viruses which are deadly pathogens to humans. Melatonin, a low-molecular-weight antioxidant, anti-inflammatory, and immunoregulatory agent, has the capacity to increase the tolerance of hosts against pathogen invasion. Since mitochondria participate in the innate immune response against virus infection, one of the mechanisms of melatonin is to preserve the structural and functional integrity of mitochondria. Melatonin is synthesized and accumulates in mitochondria in high concentrations. The high amounts are essential for maintaining the mitochondrial membrane potential, mtPTP inhibition, ROS balance, respiratory metabolism and ATP production. Conversely, melatonin deficiency favors cytochrome C release and the leakage of DAMP molecules, such as mtDNA and cardiolipin, that initiate apoptosis or/and inflammation. In addition, low melatonin allows the cell to switch from respiratory metabolism to anaerobic glycolysis. Anaerobic metabolism facilitates virus replication and spreading. It appears that viruses prefer to target melatonin synthesis by manipulating mitochondrial functions. One viral strategy is to damage the mitochondrial respiratory metabolism, which forces the cell to depend on anaerobic glycolysis. Glycolysis cannot generate acetyl-coenzyme A which is a cofactor of melatonin synthesis. Thus, viral infection will further reduce melatonin production. This will make things worse in individuals whose mitochondria are already compromised. These individuals include elderly people, some males, individuals with hypertension, obesity, cardiovascular disorders, and diabetes. This is reflected by the high severity and mortality in these subpopulations when infected by SARS-CoV-2. Hence, we recommend high dose melatonin supplementation in patients with compromised mitochondria to reduce severity and mortality after deadly viral infections, including that by SARS-CoV-2. If this strategy will be followed, it will allow for rapid reactions of the health system when new threats are emerging by as yet unknown viruses.

## Figures and Tables

**Figure 1 molecules-25-04410-f001:**
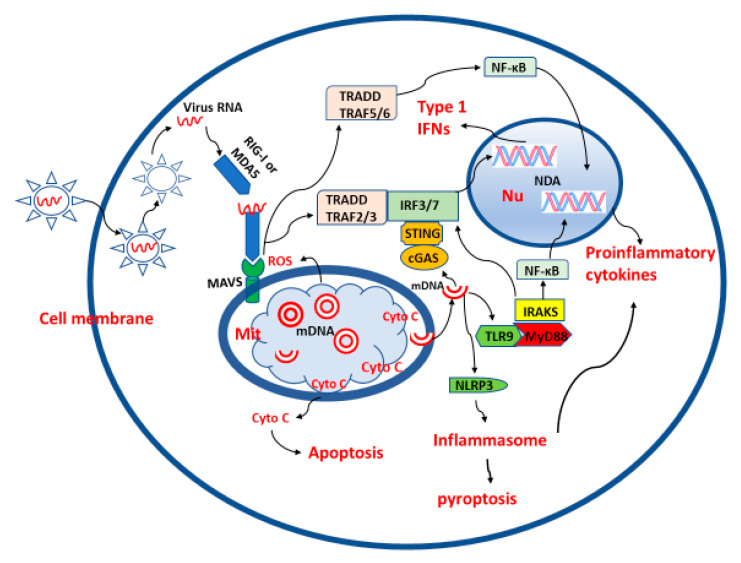
The roles of mitochondria in virus infection. For avoiding unnecessary complexity, the processes of virus uptake, passage through membranous compartments, release and disassembly of nucleocapsid, the respective representation has been strongly simplified. MIT: mitochondria, Nu: nucleus, red cycles: mitochondrial circular double strand DNA (mDNA), semi red cycles: broken mitochondrial double strand DNA, Double helix: nuclei DNA, Cyto C: cytochrome C, red wave lines: single strand viral RNA, the arrows indicate the directions of the links.

**Figure 2 molecules-25-04410-f002:**
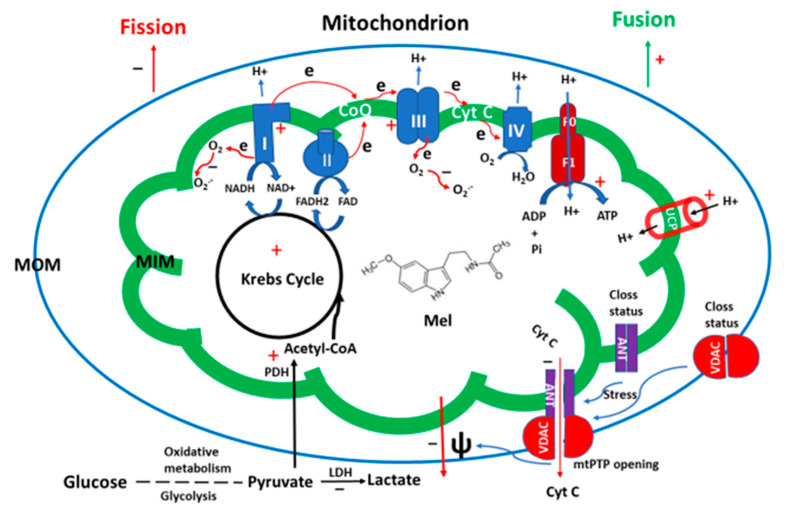
The potential effects of melatonin on mitochondria. MOM: mitochondrial outer membrane, MIM: mitochondrial inner membrane, ψ: mitochondrial membrane potential, UCP: uncoupling protein, mPTP: mitochondrial permeability transition pore, Cyt C: cytochrome C, VDAC: voltage dependent anion channel, ANT: adenine nucleotide translocase or ADP/ATP carrier, LDH: lactate dehydrogenase, PDH: pyruvate dehydrogenase, Mel: melatonin, red +: stimulating effects of melatonin, black −: suppressive effects of melatonin.

**Figure 3 molecules-25-04410-f003:**
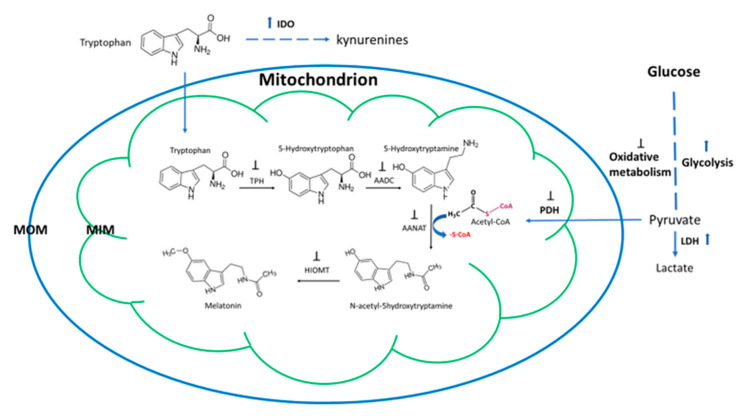
The potential mechanisms of virus to attack mitochondrial melatonin synthetic pathway. MOM: mitochondrial outer membrane, MIM: mitochondrial inner membrane, IDO: indolamine 2,3 dioxygenase, LDH, lactate dehydrogenase, PDH: pyruvate dehydrogenase, AADC: aromatic amino acid decarboxylase, AANAT: arylalkylamine *N*-acetyltransferase, HIOMT: hydroxyindole-*O*-methyltransferase, also known as *N*-acetylserotonin *O*-methyltransferase (ASMT). Black upward arrows: stimulating activities of the virus, ┴: suppressive activities of virus.

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
