# Peer review of "Targeting Host Defense System and Rescuing Compromised Mitochondria to Increase Tolerance against Pathogens by Melatonin May Impact Outcome of Deadly Virus Infection Pertinent to COVID-19"

_molecules, 2020, doi:10.3390/molecules25194410_

Round 1

Reviewer 1 Report

The manuscript „Targeting host defense system and rescuing compromised mitochondria to increase tolerance against pathogens by melatonin may impact outcome against pathogens by melatonin may impact outcome of deadly virus infection pertinent to COVID-19“ deals with a very attractive topic of the role of mitochondria in the processes of infection disease. The authors present the consideration on the potential dual strategy in fighting against viral diseases – either targeting the pathogens or targeting the host. To increase the host tolerance against pathogens seems to be the choice of defense developed in evolution giving the broader defense potential against variable infections. The strategy targeted on the protection of various functions of mitochondria including its role in the defense against viral infection seems to be a neglected but hopeful strategy. Viral infections modify the energy metabolism of mitochondria attenuating thus the synthesis of mitochondrial melatonin, limiting thus melatonin´s antioxidant and anti-inflammatory potential. The authors suggest that in conditions with already compromised mitochondrial function and reduced melatonin synthesis, such as higher age or cardiovascular pathologies, the viral infection including SARS-CoV-2 virus or influenza, may further deteriorate mitochondrial function including melatonin production. Bearing these facts in mind the authors suggest that melatonin supplementation could improve the defense and prognosis of host against viral infectious diseases.

In general, this is an attractive and very well written review, presented clearly and complexly, with discernible clinical implication. The abstract is informative and insights presented in the main body are up to date, and the individual parts are well organized, logically written and complementary. The conclusion strongly supports the suggestion to use melatonin as a therapeutic means in the battle against COVID-19. This message is rather valuable especially in the times of the lack of effective treatment and vaccination against SARS-CoV-2, and because of great safety and availability of melatonin and its potential to be used in pandemics of other viral diseases, which might emerge.

I have only minor comments:

- page 9, the authors might consider mentioning that melatonin seems to have the sympatholytic effect, which may also contribute to the improvement of energy metabolism

- recently, a short paper was published, which is in line with consideration of the authors regarding the attenuated melatonin production as a potential unifying mechanism of the worse prognosis of COVID-19 in elderly and cardiovascular pathologies (Is melatonin deficiency a unifying pathomechanisms of high risk patients with COVID-19? Life Sci 2020, 11790. DOI: 10.1016/j.lfs.2020.117902).

-the names of authors, in a number of references, are given as initials- e.g. references No. 4, 6, 8, 53, 56 and several others

Author Response

Thanks for reviewing the paper. The suggested reference has been cited.

Reviewer 2 Report

The review by Tan and Hardeland encloses comprehensive set of knowledge with regard to impact of melatonin and its targeting action against COVID-19; however, the reviewer recommend to add the latest citations which enrich the value of the article.

Introduction:

Line 86: Please add selected references describing antioxidant and anti-inflammatory properties of melatonin as follows:

Janjetovic et al., Sci. Rep. 7 (2017) 1274

Kleszczyński et al., J. Pineal Res. 58 (2015) 117-126

Kleszczyński et al., Int. J. Mol. Sci. 19 (2018) pii: E3786

Please add latest review describing melatonin as potential molecule for treatment of COVID-19:

Nutrients 2020; 12(9):E2561. doi:10.3390/nu12092561

Clinical Trials for Use of Melatonin to Fight against COVID-19 Are Urgently Needed

Author Response

Thanks for reviewing this manuscript. All references you suggested have been cited.

Reviewer 3 Report

This review manuscript titled “Targeting host defense system and rescuing compromised mitochondria to increase tolerance against pathogens by melatonin may impact outcome of deadly virus infection pertinent to COVID-19” is an interesting article that brings extra insight into the effects of melatonin against COVID-19. This manuscript is clearly the product of great effort on the part of its authors and the technical performance is clear-cut. However, the following point should be considered:

  • SARS-CoV-2 can spread into the brain dysregulating mitochondrial metabolism and cause neurological damage. It should be included briefly in the text according to neuroinvasive capacity of SARS-CoV-2 and how melatonin combats it. Then, would be the mitochondria a key target to counteract the neuroinvasion of SARS-CoV-2? The role of melatonin against the neuroinvasion of coronavirus disease is discussed by Romero et al. 2020. “Coronavirus Disease 2019 (COVID-19) and its neuroinvasive capacity: Is it time for melatonin?”. Cellular and Molecular Neurobiology DOI 10.1007/s10571-020-00938-8).

Author Response

Thanks for reviewing the paper. Your suggestion to mention the neuroinvasive activity of the SARS-CoV2 and its relation to mitochondria has been adopted and the related references have been cited.